# A Novel Multi-View Ensemble Learning Architecture to Improve the Structured Text Classification

**Carlos Adriano Gonçalves** [1,2,3,†], **Adrián Seara Vieira** [1,2,3,†], **Célia Talma Gonçalves** [4,5,‡], **Rui Camacho** [6,7,‡], **Eva Lorenzo Iglesias** [1,2,3,*,‡] and **Lourdes Borrajo Diz** [1,2,3,†]

1   Computer Science Department, University of Vigo, 36310 Vigo, Spain; coliveira@uvigo.es (C.A.G.); adrseara@uvigo.es (A.S.V.); lborrajo@uvigo.es (L.B.D.)
2   CINBIO—Biomedical Research Centre, University of Vigo, 36310 Vigo, Spain
3   SING Research Group, Galicia Sur Health Research Institute (IIS Galicia Sur) SERGAS-UVIGO, 36310 Vigo, Spain
4   CEOS.PP—Instituto Superior de Contabilidade e Administracao do Porto ISCAP, Polytechnique of Porto, Rua Jaime Lopes Amorim, s/n, S. Mamede de Infesta, 4465-004 Porto, Portugal; celia@iscap.ipp.pt
5   LIACC, Campus da FEUP, Rua Dr. Roberto Frias, 4200-465 Porto, Portugal
6   Faculdade de Engenharia da Universidade do Porto, Rua Dr. Roberto Frias s/n, 4200-465 Porto, Portugal; rcamacho@fe.up.pt
7   LIAAD-INESC TEC, Campus da FEUP, Rua Dr. Roberto Frias, 4200-465 Porto, Portugal
*   Correspondence: eva@uvigo.es
†   Current address: Escuela Superior de Ingeniería Informática, Campus Universitario As Lagoas, 32004 Ourense, Spain.
‡   These authors contributed equally to this work.

**Abstract:** Multi-view ensemble learning exploits the information of data views. To test its efficiency for full text classification, a technique has been implemented where the views correspond to the document sections. For classification and prediction, we use a stacking generalization based on the idea that different learning algorithms provide complementary explanations of the data. The present study implements the stacking approach using support vector machine algorithms as the baseline and a C4.5 implementation as the meta-learner. Views are created with OHSUMED biomedical full text documents. Experimental results lead to the sustained conclusion that the application of multi-view techniques to full texts significantly improves the task of text classification, providing a significant contribution for the biomedical text mining research. We also have evidence to conclude that enriched datasets with text from certain sections are better than using only titles and abstracts.

**Keywords:** full text classification; stacking; multi-view ensemble learning; ensemble methods; OHSUMED corpus

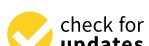

## 1. Introduction

In recent years, with the advancement of information technology, many new applications, such as social networks and e-Commerce, have emerged as hubs for information gathering and dissemination. The majority of the information generated and shared in the Internet is in the form of text data, for example, reports, scientific articles, tweets, product reviews, etc.

Text classification has an important role to handle and organize text data in real-world scenarios, such as classifying webpages or documents, user sentiment analysis for social network multimedia, spam email filtering, disseminating information, document genre identification, recommendation systems, etc.

Many different machine learning methods are being used to text classification, including support vector machines (SVM), logistic regression, boosting, Naive Bayes, nearest neighbor (kNN), and neural-networks-based systems.

For the classification of scientific texts, these methods are generally applied only on titles and abstracts. However, users searching full texts are more likely to find relevant articles than when searching only titles and abstracts. This finding affirms the value of full text collections for text retrieval and provides a starting point for future work in exploring algorithms that take advantage of rapidly growing digital archives.

In this paper, we show the advantages of using ensemble learning techniques and multi-view models to full text classification. The primary aim is to take advantage of the organization in structured documents in order to build different views to represent each instance. In this way, an ensemble of classifiers can be created by training a specific classifier per view. To obtain views of data, we propose the use of text sections.

Another focus of the research is to address the comparison between full text classification and only titles and abstracts as well as the use of specific sections to conclude their significance in the scientific article classification process.

The rest of the paper is organized as follows. Section 2 reviews the concepts of ensemble classifiers and multi-view ensemble learning. Section 3 introduces related works to the research scope of this paper. In Section 4, we present a novel multi-view ensemble learning scheme for structured text classification. Section 5 demonstrates the use of the technique to classify a biomedical full text corpus. Sections 6 and 7 show and discuss the results, respectively. Finally, Section 8 concludes the paper by summarizing the main contributions and presenting future work to improve the model.

## 2. Theoretical Background

### 2.1. Ensemble Classifiers

Unlike traditional classification techniques, ensemble classifiers, also called multiple classifiers, are becoming very popular in the scientific field because they take into account all valid hypotheses by combining the ensemble predictions. Ensembles have been shown to be an efficient way of improving predictive accuracy or/and decomposing a complex, difficult learning problem into easier sub-problems [1]. They also solve overfitting problems and obtain good results with little training data.

An ensemble classifier is a set of classifiers whose individual predictions are combined, with the objective of improving accuracy over an average classifier. Since it is not known a priori which classifier is best for a particular problem, an ensemble reduces the risk of selecting a poorly performing classifier [2].

Usually, the combination of classifiers is done by the voting scheme so that each classifier casts one vote. The example is classified into the class that obtains the highest number of votes. However, this technique based on majority voting works well when all the combined models have acceptable accuracy, and that cannot be assured for all classifiers.

An alternative to majority voting is the introduction of a meta-learning phase to learn how to best combine the predictions from the first phase [2,3]. Meta-learning focuses on predicting the right algorithm for a particular problem based on the characteristics of the dataset or based on the performance of other simpler learning algorithms [4].

Stacking is probably the most popular meta-learning technique [2,5]. It is a general procedure where a meta-learner is trained to combine the base learners. The stacking algorithm has two phases. First, the base algorithms are trained to obtain predictions based on a training dataset. Subsequently, from the results obtained by the base algorithms, meta-level training data with the same classes as the original dataset are obtained, using a procedure similar to k-fold cross-validation [6].

### 2.2. Multi-View Ensemble Learning

Multi-view algorithms take advantage of datasets that have a natural separation of their features or can be described using different "kinds" (views) of information [7]. The diversity of the various views provides insight for learning from multiple views of the data.

The views may be obtained from multiple data sources or different feature sets. A prominent example is web pages, which can be classified based on their content as well as on the hyperlinks [8].

Multi-view ensemble learning (MEL) has the potential to explore the use of features in multiple views of the data for the classification task [9].

## 3. Related Works

In reviewing the literature on multi-view ensemble learning, we find it has been applied to manage and organize text data in different scenarios.

Bai and Wang [10] use the information of MEL to identify new malware in a current anti-virus. They propose two malware detection schemes that incorporate three-feature views (byte n-grams, opcode n-grams, and format information) using different ensemble learning mechanisms to arrive at detecting new and unknown malware. Experimental results show that the proposed schemes do enhance the performance of new malware detection and the generalization performance of the learning model with respect to single-view feature methods.

Cuzzocrea et al. [11] propose a multi-view ensemble learning approach, which exploits the multi-dimensional nature of log events of a business process and implements a context- and probability-aware stacking method for combining base model predictions. It is based on the idea of training multiple base classifiers on different pattern-based views of the log and then combining them via a stacking-oriented meta-learning scheme. Tests on a real-life log confirmed the validity of the approach and its capability to achieve compelling performances.

Liu et al. [12] propose a novel method for product defect identification from online forums that incorporates contextual features derived from replies and uses a multi-view ensemble learning method specifically tailored to the problem on hand. Each base classifier is based on one of four categories of features (linguistic features, social features, distinctive terms, and contextual features), which are considered different views of the data. In addition, another base classifier is trained based on a subset of features using a correlation-based feature selection method. Finally, the five base classifiers are combined using a logistic regression model. A case study in the automotive industry demonstrates the utilities of the novel method.

In the text mining area, especially when dealing with issues such as multi-label learning and high-dimensional data processing, MEL can effectively better model performance. Recently, Frag et al. [13] created a new ensemble method for multi-view text clustering that exploits different text representations in order to produce more accurate and high-quality clustering. Views are generated using the Vector Space, the Latent Dirichlet Allocation and the Skip-gram models. Once the views are generated using the three different representation models, the K-means algorithm is applied on each of the matrices corresponding to each view, and the combination is based on two different ensemble techniques (Cluster-Based Similariy Partitioning and Pairwise Dissimilarity Matrix). The conducted experimentation to trends detection from Twitter shows that in comparison to single view-based clustering, using multiple views improves the clustering quality.

Finally, Ye et al. [14] apply MEL to fuse the information contained in different features for better microblog sentiment classification. Features are created by different text representation methods (the Chi-square statistic, the emoticon space mapping and the emotional element statistical methods) during model construction, which is useful to comprehensively identify the sentiment from multiple views. The different raw features are combined in different ways to construct the basic classifiers. To better integrate different basic classifiers, a local fusion stage and a global fusion stage are considered. In the local fusion stage, the basic classifiers are combined into five classifier groups. In the global fusion stage, these classifier groups are further integrated to make more accurate and comprehensive predictions. To evaluate the performance of the proposed method, two public Chinese microblog benchmark datasets are applied. The experimental results demonstrate that

the proposed method enhances the performance of microblog sentiment classification by effectively fusing multi-view information.

In all of the presented methods, the views correspond to different sources of the same document; i.e., for scientific papers, a view can refer to the actual text while another view refers to the citations. However, deriving multiple views from the same text document has not been tackled in the literature with the exception of those mentioned above [13,14]. Therefore, for the same document, considering different representations of text as views can improve the classification performance. The proposed approach combines classification results obtained individually from each view on the basis of a consensus, where each view is a section text.

## 4. Material and Methods

Figure 1 shows the training workflow of the proposed multi-view ensemble learning architecture, consisting of two main steps: a view generation phase and an ensemble training phase.

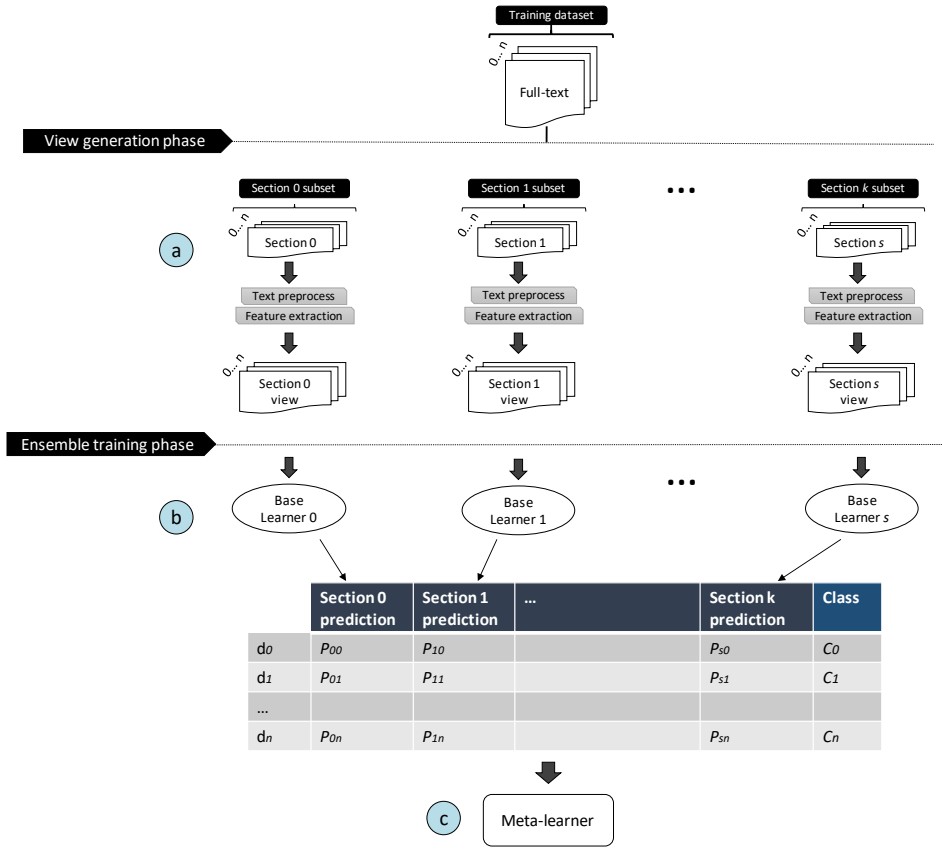

**Figure 1.** Proposed multi-view ensemble learning architecture: (**a**) view generation phase, (**b**) ensemble training phase, (**c**) metalearning.

In the view generation phase, each document of the initial training set is divided into $s$ sections, where $s$ is the number of sections in which the texts can be structured. This first step ends up creating $s$ training subsets, each one containing texts corresponding to a concrete section of the documents. Then, each section subset is preprocessed in order to build a section-dependent representation for the texts. This representation is called **section view** and it consists of a feature space to represent a text instance with information extracted from that specific section.

Ensemble training phase is based on the stacking paradigm. The idea of this method is, as mentioned, to train different base models (also called **weak learners**) and combine them by training a meta-learner to classify new text instances based on the multiple predictions

obtained by the base learners. As shown in Figure 1, we use the section views obtained in the first phase to train the base learners.

Specifically, a base learner is trained with each view in order to build the meta-leaner training dataset. This dataset contains an array for every text instance in the original training set where elements indicate the predictions made by the base learners. The process of building this dataset of predictions is depicted in Figure 2.

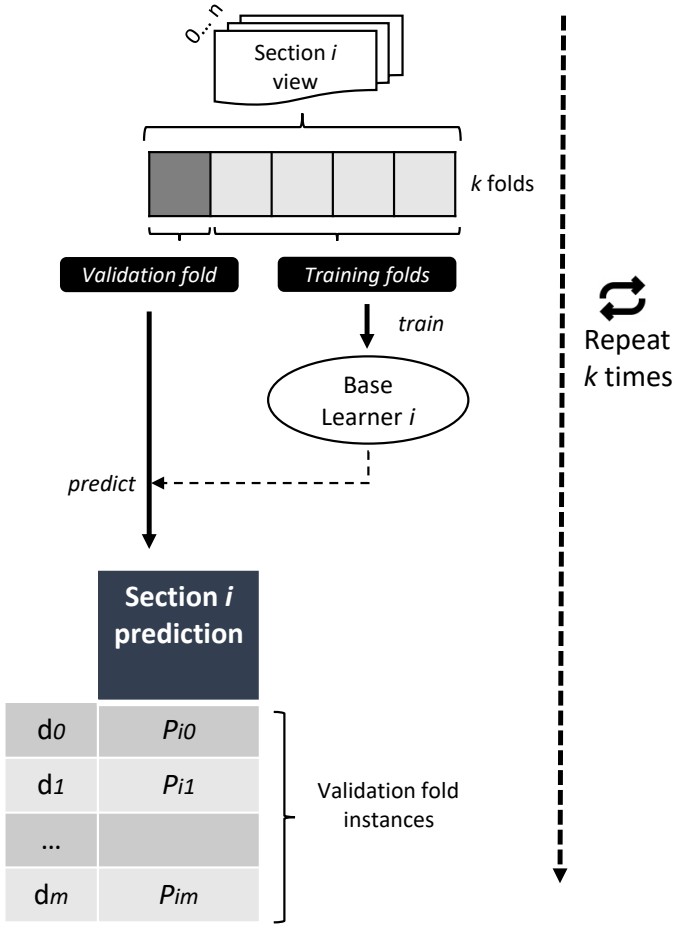

**Figure 2.** Meta-learner training dataset creation.

The training subset corresponding to each section is divided in *k* folds. One fold is considered the validation set, and the base learner is trained with the remaining folds. The trained classifier is then used to predict the text instances in the validation set and the process is repeated *k* times, considering a different validation fold in each iteration.

Once the training process has finished, the system will be able to classify new instances by following similar steps. Whenever a new text instance needs to be classified, it is divided into its corresponding sections, and a view per section is created. Each base learner emits an output class taking into account the view of the text instance that it was trained for. Then, the meta-classifier decides the final class by classifying the vector formed by the outputs of every base learner.

It is important to note that the proposed multi-view ensemble architecture has three adaptable parts that may be configured or adjusted depending on the input data. As it can be seen in Figure 1, these are:

1. Figure 1a: The text pre-processing techniques that are applied in order to create each section view. Different pre-processing methods may be applied to different sections.
2. Figure 1b: The classifying algorithms that are used as base learners. Different classifiers may be used for different sections.

3. Figure 1c:The classifying algorithm used as a meta-classifier.

## 5. Experiments

The proposed architecture is able to classify any data that can be transformed into structured text instances. In order to test its performance, the system is used to classify Medline full text scientific documents.

Figure 3 shows the complete configuration of the proposed architecture used for the experiments. It has two phases that need to be defined before training: the view generation phase and the ensemble training phase.

Previously, we have to obtain a reliable and pre-classified full text corpus, as explained below.

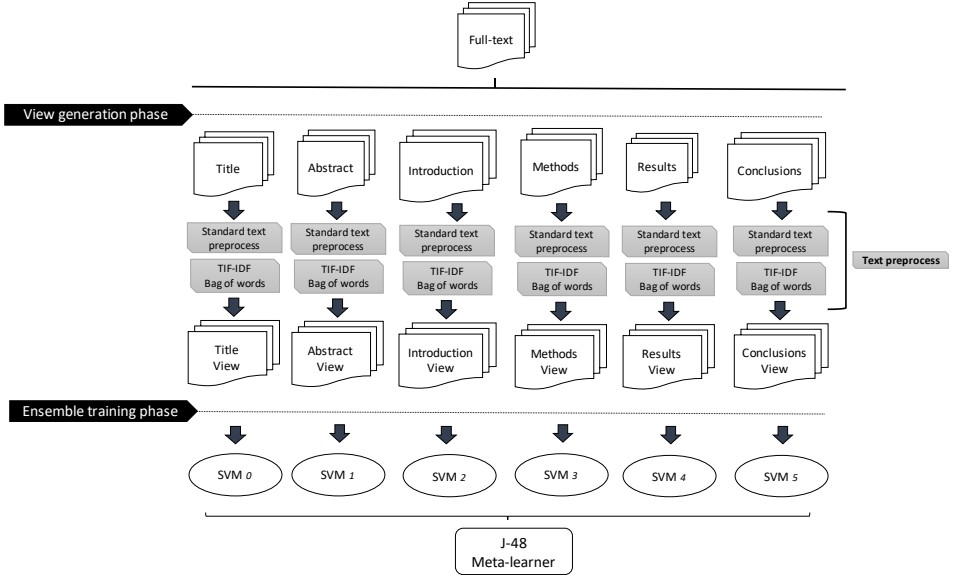

**Figure 3.** Experimental setup.

### 5.1. Dataset Construction

For the purpose of this study, we use a corpus based on OHSUMED [15]. OHSUMED is composed of 34, 389 MEDLINE documents that contain Title, Abstract, MeSH terms, author, source and publication type of biomedical articles published between 1988 and 1991.

Each document of OHSUMED has one or more associated categories (from 26 diseases categories). To carry out a binary classification, we select one of these categories as relevant and consider the others as non-relevant. If a document has assigned two or more categories and one of them is the one considered relevant, then the document is considered relevant and is excluded from the set of non-relevant documents.

For example, in order to build a corpus for the C14 Cardiovascular Diseases category, we select documents that belong to the C14 category as relevant. Then, from the common bag of non-relevant categories, all the possible documents categorized as "Cardiovascular Diseases" are removed. The resultant set is taken as the non-relevant set of documents. In this way, the number of relevant and non-relevant documents on each corpus is shown in Table 1.

Note that C21 and C24 categories were discarded because they have only 1 and 17 relevant documents, respectively.

**Table 1.** Number of relevant and non-relevant documents of OHSUMED that we have assembled for each category.

| Dataset | Description | Rel. | Non Rel. |
|---|---|---|---|
| C01 | Bacterial Infections and Mycoses | 417 | 13,625 |
| C02 | Virus Diseases | 1178 | 13,080 |
| C03 | Parasitic Diseases | 51 | 13,884 |
| C04 | Neoplasms | 5537 | 8789 |
| C05 | Musculoskeletal | 51 | 13,884 |
| C06 | Digestive System | 1662 | 12,484 |
| C07 | Stomatognathic | 145 | 13,372 |
| C08 | Respiratory Tract | 857 | 13,184 |
| C09 | Otorhinolaryngologic | 215 | 13,845 |
| C10 | Nervous System | 2780 | 11,394 |
| C11 | Eye Diseases | 392 | 13,699 |
| C12 | Urologic and Male Genital Diseases | 1196 | 12,985 |
| C13 | Female Genital Diseases and Pregnancy Complications | 1136 | 12,954 |
| C14 | Cardiovascular Diseases | 2532 | 11,792 |
| C15 | Hemic and Lymphatic | 450 | 13,756 |
| C16 | Neonatal Diseases and Abnormalities | 469 | 13,753 |
| C17 | Skin and Connective Tissue | 1227 | 13,072 |
| C18 | Nutritional and Metabolic | 1043 | 13,267 |
| C19 | Endocrine Diseases | 772 | 13,415 |
| C20 | Immunologic Diseases | 1721 | 12,536 |
| C22 | Animal Diseases | 76 | 13,964 |
| C23 | Pathological Conditions, Signs and Symptoms | 7,191 | 7,136 |
| C25 | Chemically-Induced Disorders | 174 | 13,995 |
| C26 | Wounds and Injuries | 247 | 13,949 |

As OHSUMED only contains the title and abstract of the documents, we downloaded a full text corpus available at PubMed/NCBI (459,009 documents in total). The PubMed tool provides access to references and abstracts on life sciences and biomedical topics. Most of these documents are manually annotated by health experts with the MeSH Heading descriptors under 16 major categories, which facilitates the search for specific biomedical-related topics.

To obtain the MeSH terms, we downloaded the 2017 MeSH trees from NCBI. MEDLINE MeSH Headings are mapped with OHSUMED categories through the MeSH terms associated.

The documents were filtered by the MeSH classes (Medical Subject Headings), the National Library of Medicine (NLM) controlled vocabulary thesaurus used for indexing PubMed articles, and the corresponding full text documents were obtained from the NCBI PubMed Central (PMC) repositories.

Another important issue to mention is that all the MEDLINE scientific full text documents contained in the corpus have a common structure, which we aggregated according to the following sections: Title, Abstract, Introduction, Methods (Materials and Methods, Methods, Experimental Procedures), Results (Results, Discussion, Results and Discussion) and Conclusions.

Finally, we obtained an OHSUMED-based full text corpus. A more detailed description of the full text document corpus creation process is available at [16,17].

### 5.2. View Generation Phase: Text Pre-Processing

Once the full text corpus has been created, it is necessary to reduce the number of features in order to be manageable for the classifiers.

We now briefly detail each of the pre-processing techniques applied that highly reduce the number of attributes in the corpus: Named Entity Recognition (NER) identification, stopwords removal, synonyms replacement, word validation using dictionaries and ontologies, and stemming to the full text documents. We previously evaluated them in [18]:

- Named Entity Recognition (NER): Named Entity Recognition (NER) is the task of identifying terms that mention a known entity in the text. Entities typically fall into a pre-defined set of categories such as person, location, organization, etc. For the purpose of our work, we are interested in identifying entities from the Life Sciences such as proteins, genes, etc. For this reason, we used the Biomedical Named Entity Recognition tool called ABNER [19];

- Special characters removal: punctuation, digits and some special characters (such as ";"; ":"; "!"; "?"; "0"; "[" or "]") are removed;

- Tokenization: splits the document sections into tokens, e.g., terms or attributes;

- Stopwords removal: It removes words that are meaningless such as articles, conjunctions and prepositions (e.g., "a", "the", "at"). We used a list of 659 stopwords to be identified and removed from the documents;

- Dictionary Validation: A term is considered valid if it appears in a dictionary. We gathered several dictionaries for common English terms, such as ispell https://www.cs.hmc.edu/~geoff/ispell-dictionaries.html (accessed on 29 March 2022) and WordNet http://wordnet.princeton.edu/ (accessed on 29 March 2022) [20]. For biological and medical terms, we used BioLexicon [21], the Hosford Medical Terms Dictionary and Gene Ontology (GO) http://www.geneontology.org/ (accessed on 29 March 2022);

- Synonyms handling: using the WordNet (an English lexical database) for regular English ("non technical" words) and Gene Ontology for technical terms;

- Stemming: It is the process of removing inflectional affixes of words, thus reducing the words to their root. We used the Porter Stemmer algorithm [22] to normalize several terms variants into the same form and to reduce the number of terms;

- Bag of Words (BoW): It is the traditional representation of a document corpus. A document–term matrix is used, where each row represents a document from the corpus and each column represents a word of the vocabulary of the corpus. The weight calculation uses the normalized frequencies of the words that is given by the Term Frequency-Inverse Document Frequency (TF-IDF) [23].

After applying the pre-processing techniques to the original full text corpus, we obtain a section-term matrix containing weighted terms for each of the text sections: Title, Abstract, Introduction, Methods, Results and Conclusions.

Table 2 shows an example of term frequencies by section in a PubMed document. The dividend indicates the number of occurrences of the term in the section of the document analyzed, while the divisor is the number of occurrences of the term in the section in all documents of the corpus.

Finally, Table 3 summarizes the number of terms per section for the OHSUMED corpus.

**Table 2.** Example of term frequencies by section, for some terms of the document with pmid identificator 18459944. Divisors represent the total number of occurrences of each term in each section for all documents of the corpus.

| Terms | Title | Abstract | Introduction | Methods | Results | Conclusions |
|---|---|---|---|---|---|---|
| angiotonin | 0/31 | 1/104 | 0/506 | 2/231 | 20/972 | 0/122 |
| collagen | 0/14 | 0/327 | 0/1878 | 2/2040 | 0/9248 | 0/594 |
| diabet | 0/214 | 4/1493 | 5/5573 | 0/6299 | 42/22,692 | 0/2463 |
| hypertens | 1/136 | 7/914 | 6/3319 | 6/3664 | 164/14,274 | 0/1203 |
| insulin | 0/107 | 0/925 | 0/4511 | 4/3955 | 6/16,962 | 0/1001 |
| kidnei | 1/96 | 2/669 | 4/2654 | 28/3152 | 58/11,594 | 0/636 |
| methanol | 0/0 | 0/2 | 0/38 | 4/1696 | 0/310 | 0/132 |
| pathogenesi | 0/28 | 0/653 | 0/2509 | 0/213 | 2/4402 | 0/377 |

**Table 3.** Number of terms by section.

| Section | #Terms |
|---|---|
| Title | 4798 |
| Abstract | 8822 |
| Introduction | 14,130 |
| Methods | 15,948 |
| Results | 19,255 |
| Conclusions | 11,257 |

### 5.3. Ensemble Training Phase: Base Learners and Meta-Learner

The main objective of the **ensemble training** phase is to apply a stacking algorithm that can combine several machine learning methods or datasets.

In this study, the ensemble learning model combines several classifiers in two levels. The first level runs the base classifier (a support vector machine—WEKA Functions SMO [24] using a Radial Basis Function kernel with default parameters) over the six mentioned sections. SVMs have proven to be one of the best text classification models in numerous studies.

Internally, the stacking algorithm gathers the results of the previous level and creates a new dataset for the final layer. Decision trees have proven to be a robust meta-learner in a stacking paradigm [25,26]. In our experiments, the C4.5 decision tree algorithm—WEKA Trees J48 [27] implementation is used as a meta-learner. The combination of the outputs of the six sections (Title, Abstract, Introduction, Methods, Results and Conclusions) aims to achieve better results.

It is important to guarantee the reliability and robustness of the developed model, e.g., its accuracy and classification success when the model is applied to new unseen documents. The selection of the base classifiers and the meta-learner has been done after conducting an initial set of experiments with the OHSUMED full text corpora, outperforming other classifiers in this step such as Naive Bayes or SVM.

In order to reduce bias and variance in evaluation, we perform a ten-fold cross-validation with the proposed architecture for each one of the 24 OHSUMED corpora chosen.

In addition, the same ten-fold cross-validation is executed following a standard text classification process, using only Title and Abstract texts with a single SVM. This is the reference point in order to compare the results achieved by the proposed architecture with classical single-view classifiers.

### 5.4. Kappa Statistics and Statistical Significance

The metric used for the evaluation is the Kappa value [28–30]. Kappa represents the value of Cohen's Kappa coefficients, which is a statistical measure that determines the agreement between different classifiers (see Table 4).

**Table 4.** Kappa statistics.

| Kappa Agreement | |
|---|---|
| <0 | Less than chance |
| 0.01–0.20 | Slight |
| 0.21–0.40 | Fair |
| 0.41–0.60 | Moderate |
| 0.61–0.80 | Substantial |
| 0.81–0.99 | Almost perfect |

This measure takes into account the possibility of casual successes, and it can take values between −1 and 1, in which the negative values indicate that there is no agreement, and the values between 0 and 1 indicate the level of existing agreement. Between 0.01 and 0.20 is a slight agreement, between 0.21 and 0.40 is fair agreement, between 0.41 and 0.60 is

moderate agreement, between 0.61 and 0.80 is substantial agreement, and between 0.81 and 1.0 is perfect agreement. Given these values, the larger the agreement, the more reliable the results of the classifiers.

The WEKA tool uses the Corrected Resample *t*-test [31] as a statistical significance of a pair-wise comparison of schemes. According to the study [31], this approach performs better than the standard t-test, and this technique is used as default by the WEKA tool used in this study.

## 6. Results and Discussion

To evaluate the effectiveness of the proposed multi-view ensemble learning method to structured text classification, two different studies were carried out. Firstly, a comparison was between classical text classification on Title and Abstract and classification using the proposed multi-view ensemble learning technique on full text sections. Subsequently, a study was carried out on the efficiency of the model with respect to full text classification techniques using a single classifier. They are explained in detail below.

### 6.1. Comparing Title–Abstract Classification vs. Full Text Classification

This first study was carried out in order to analyze the advantages of classifying texts using all their content through a multi-view approach as opposed to using only the Title and Abstract.

Figure 4 shows the number of OHSUMED datasets of Table 1 where each method was statistically better: Stacking (multi-view ensemble learning on text sections), Title–Abstract (single SVM classifier on Title-Abstract), or None if there is no statistical difference.

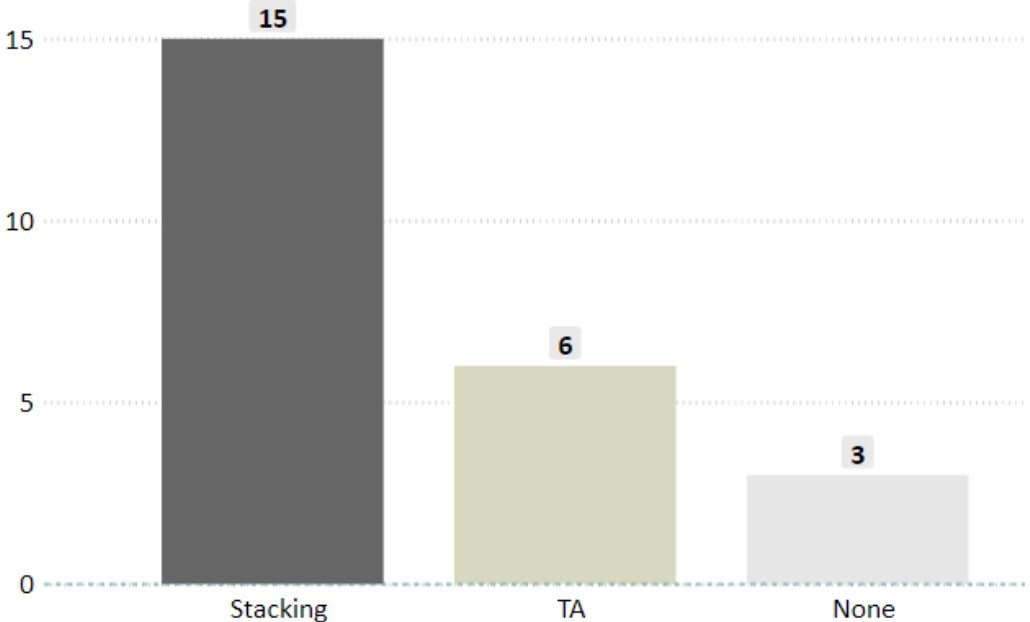

**Figure 4.** Number of OHSUMED datasets (Table 1) where each method (Stacking or Title-Abstract TA) was statistically better, or None if there is no statistical difference.

Figure 5 displays the Kappa values achieved by the OHSUMED datasets where stacking is the best statistically tested technique.

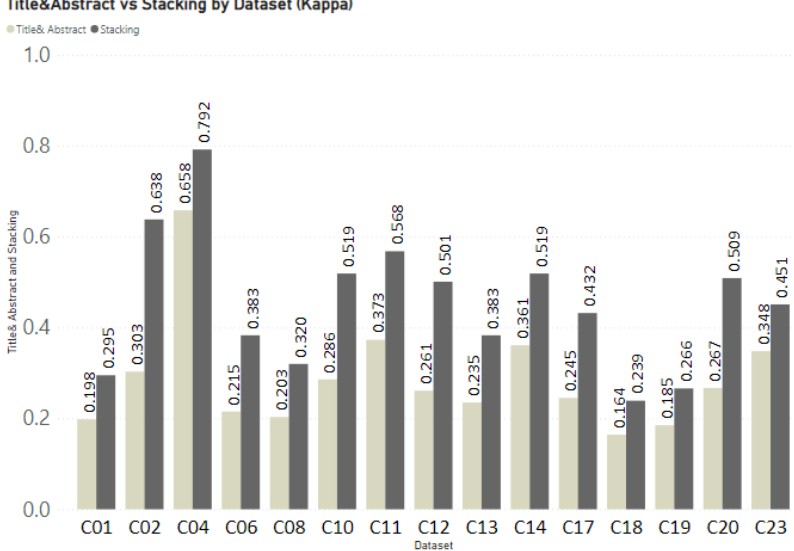

**Figure 5.** Kappa values of the datasets where stacking is the best statistically tested technique.

Figure 6 shows the Kappa values achieved by the OHSUMED datasets where the classification of Title–Abstract using a single SVM classifier (TA) is the best statistically tested technique.

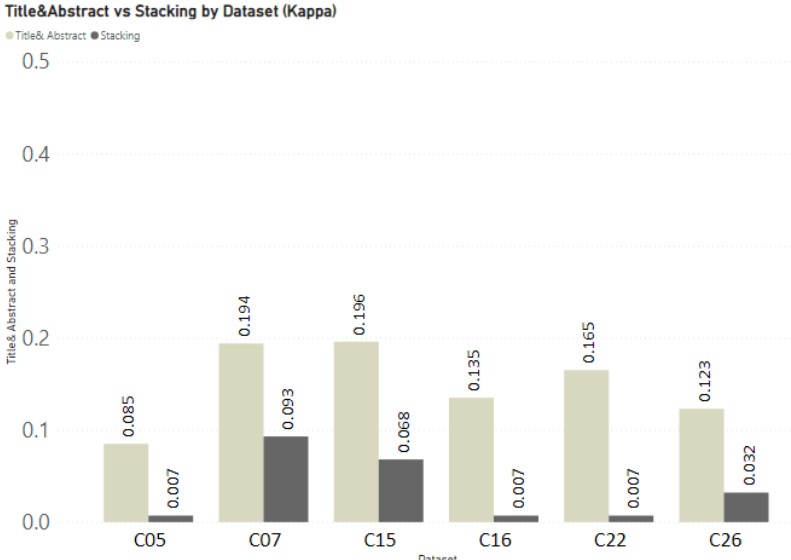

**Figure 6.** Kappa values of the OHSUMED datasets where Title–Abstract (TA) is the best statistically tested technique.

Finally, Figure 7 provides the Kappa values obtained by the OHSUMED datasets where the statistical difference between Stacking versus Title–Abstract models cannot be tested.

As it can be seen, the proposed stacking method statistically outperforms the standard text classification approach in 15 of the 24 tested corpora. This suggests that the usage of different text sections can improve the accuracy of the final classification process.

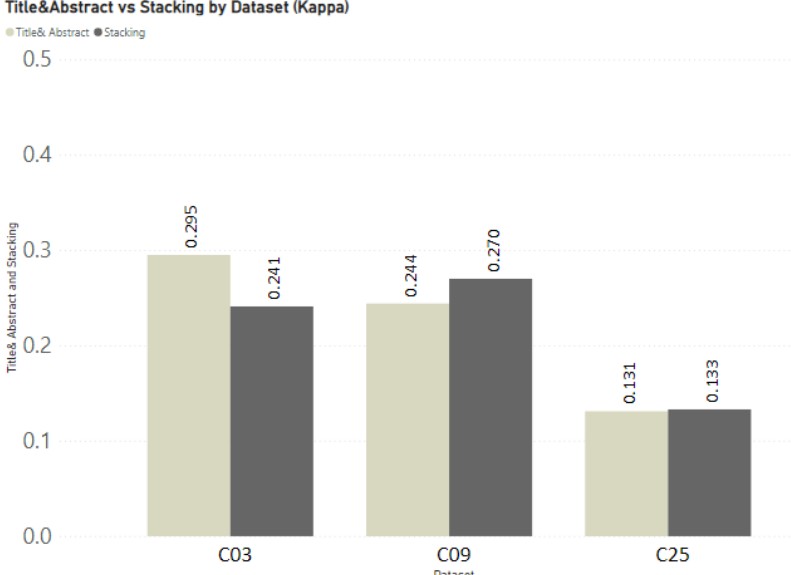

**Figure 7.** Kappa values of the OHSUMED datasets where the statistical difference between Stacking–TA models cannot be tested.

### 6.2. Comparing Full Text Classification with Stacking vs. Single Classifier

A second study was carried out to contrast the results obtained by the stacking approach versus the use of a single classifier to classify full text documents.

Specifically, five OHSUMED full text datasets were classified using the SVM implementation of Weka. Table 5 shows the Kappa values achieved by the single classification, together with those corresponding to the multi-view classification using the novel stacking approach.

**Table 5.** Kappa values obtained by the base classifiers.

| Corpus | Multi-View Full Text | Single SVM Full Text |
|--------|----------------------|----------------------|
| C01 | 0.30 | **0.43** |
| C04 | 0.79 | **0.82** |
| C06 | 0.38 | **0.54** |
| C14 | 0.52 | **0.63** |
| C20 | 0.51 | **0.62** |

As it can be seen, the standard full text classification approach outperforms the proposed stacking method in all the tested datasets. However, it should be noted that the stacking approach is a more advantageous option, since it allows the adaptation of each base learner according to the characteristics and the content of each view.

Related to the above, the results obtained by the base classifiers were analyzed in order to check which views (text sections) provided the best results. Table 6 shows the Kappa values achieved by each section separately, taking only into account the output of the base learner for each section. The values show that the Results section obtains the highest Kappa values for all the datasets analyzed, and the Conclusions section shows the lowest values in almost all experiments (except for the C18 dataset). This further supports the need of assessing the relevance of each section depending on the corpus that can lead to a better overall performance.

**Table 6.** Kappa values obtained by the base classifiers on the different views (text sections).

| Corpus | Title | Abstract | Introd. | Methods | Results | Conclusions |
|--------|-------|----------|---------|---------|---------|-------------|
| C01 | 0.13 | 0.15 | 0.17 | 0.30 | **0.38** | 0.05 |
| C02 | 0.17 | 0.26 | 0.39 | 0.55 | **0.64** | 0.09 |
| C03 | 0.26 | 0.32 | 0.22 | 0.33 | **0.34** | 0.02 |
| C04 | 0.16 | 0.61 | 0.56 | 0.59 | **0.65** | 0.10 |
| C05 | 0.06 | 0.07 | 0.11 | 0.11 | **0.18** | 0.03 |
| C06 | 0.10 | 0.17 | 0.15 | 0.37 | **0.49** | 0.05 |
| C07 | 0.16 | 0.16 | 0.13 | 0.19 | **0.23** | 0.05 |
| C08 | 0.12 | 0.17 | 0.14 | 0.34 | **0.45** | 0.06 |
| C09 | 0.20 | 0.18 | 0.25 | 0.30 | **0.34** | 0.15 |
| C10 | 0.15 | 0.24 | 0.26 | 0.45 | **0.50** | 0.11 |
| C11 | 0.21 | 0.37 | 0.23 | 0.52 | **0.55** | 0.08 |
| C12 | 0.10 | 0.25 | 0.23 | 0.38 | **0.54** | 0.06 |
| C13 | 0.12 | 0.21 | 0.22 | 0.34 | **0.44** | 0.08 |
| C14 | 0.15 | 0.31 | 0.25 | 0.45 | **0.51** | 0.15 |
| C15 | 0.17 | 0.11 | 0.13 | 0.12 | **0.24** | 0.04 |
| C16 | 0.09 | 0.09 | 0.16 | 0.09 | **0.17** | 0.03 |
| C17 | 0.10 | 0.22 | 0.24 | 0.39 | **0.52** | 0.04 |
| C18 | 0.07 | 0.12 | 0.15 | 0.28 | **0.39** | 0.09 |
| C19 | 0.09 | 0.15 | 0.16 | 0.28 | **0.43** | 0.03 |
| C20 | 0.16 | 0.21 | 0.28 | 0.45 | **0.52** | 0.07 |
| C22 | 0.10 | 0.13 | 0.10 | 0.11 | **0.20** | 0.00 |
| C23 | 0.22 | 0.35 | 0.29 | 0.32 | **0.36** | 0.07 |
| C25 | 0.13 | 0.06 | 0.12 | 0.21 | **0.32** | 0.03 |
| C26 | 0.07 | 0.13 | 0.07 | 0.18 | **0.23** | 0.03 |

## 7. Discussion

Multi-view ensemble learning exploits the information of data views. To test its efficiency for full text classification, in this work, a novel stacking technique where the views correspond to the document sections is proposed.

Experiments were performed on the OHSUMED dataset using SVM classification algorithms to base learners and C4.5 decision tree as a meta-learner.

From the experiments, we can conclude that text classification benefits more from ensemble learning methods on full text than traditional classifiers on Title and Abstract. The results are better in 15 of the 24 datasets. In five of the datasets, traditional classification on Title and Abstract achieves better results, and in three of the datasets, neither is better or worse.

With respect to full text classification using a simple model, the results obtained by our multi-view ensemble learning model are somewhat worse. However, it should be noted that it has not yet been analyzed in depth which classifiers are most suitable for each type of data source (view). In addition, each base classifier deals with a subset of the entire full text vocabulary, which can be beneficial for future improvements in each section regarding pre-processing.

This study also shows the varying importance of the different sections in the full text classification process. As can be seen in Table 6, there are sections that clearly increase the efficiency of the classification with respect to that which is commonly used on the Title and Abstract.

In any case, given that each base classifier works with one section, we believe that sections with lower accuracy are compensated for by other classifiers, thus improving the accuracy of the full text classification.

## 8. Conclusions and Future Work

The main contributions of the paper can be summarized as follows:

- We propose a novel, efficient multi-view ensemble classification scheme based on stacking. Our experimental comparison with the traditional classification indicates that the proposed scheme is better when used for text classification;
- The work contributes by providing significant benefits for the biomedical full text document mining research. To the best of our knowledge, our study is the first to use a multi-view ensemble learning schema for full text scientific document classification;
- Although the proposed classification scheme was developed based on an empirical analysis of biomedical documents, it can be applied to several other structured text corpora, including web pages, blogs, tweets, scientific text repositories or full text databases.

As a future work, it would be optimal to automatically deduce which models are the best suited. Indeed, each corpus and each section should be treated with the most appropriate algorithm (base-learner) to later select a meta-classifier that outperforms the rest. Depending on the number of features and the size of the vocabulary corresponding to each view, other types of classifiers will be tested. In addition, the use of ontologies to enrich the vocabulary in each case will also be considered.

**Author Contributions:** Conceptualization, R.C. and E.L.I.; methodology, L.B.D.; software, A.S.V. and C.A.G.; validation, C.A.G. and C.T.G.; formal analysis, R.C. and C.A.G.; investigation, L.B.D. and E.L.I.; resources, R.C.; data curation, C.A.G. and C.T.G.; writing—original draft preparation, A.S.V.; writing—review and editing, E.L.I. and A.S.V.; visualization, E.L.I.; supervision, L.B.D.; project administration, R.C. All authors have read and agreed to the published version of the manuscript.

**Funding:** This research received no external funding.

**Institutional Review Board Statement:** Not applicable.

**Informed Consent Statement:** Not applicable.

**Data Availability Statement:** Data has been presented in the main text.

**Acknowledgments:** This study was supported by MCIN/AEI/10.13039/501100011033 under the scope of the CURMIS4th project (Grant PID2020-113673RB-I00).

**Conflicts of Interest:** The authors declare no conflict of interest.

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
