# Peer review of "A Novel Multi-View Ensemble Learning Architecture to Improve the Structured Text Classification"

_information, doi:10.3390/info13060283_

Round 1
Reviewer 1 Report
The article proposes a novel multi-view ensemble learning architecture for structured text classification. The proposed methodology is well documented and the figures in the article are informative.
However, the proposed method is well-designed, the efficiency and usefulness of the proposed method need to be proven by evaluating more documents.
Main comments:
- The introduction does not give an overview of the aim of the proposed methodology and the possible application areas. Subsections “Ensemble Classifiers” and “Multi-view Ensemble Learning” present the related machine learning terms and areas at a high standard, but the aim and motivation of the research are entirely missing before these subparagraphs. Therefore, I suggest moving these subsections into a new Section (e.g. called “Theoretical background”) and inserting a brand new section (Introduction) before them containing the scope, the motivation of the research, and the possible application areas.
- The main problem of the article arises from the methodology of the comparison of the results. If I understand correctly, the efficiency of the proposed method was compared to text classification results processing only on the Title and the Abstracts of the documents. It is not fair because the proposed method could process the whole document, but the baseline method only the title and the abstract. Therefore, it would be necessary to compare the results of the multi-view ensemble method with the result of other (not multi-view) methods processing the whole text of the documents.
Minor comments:
- Related works and their efficiency should be presented in more detail.
- Please clarify the terms “multi-view semi-supervised algorithms” through an easy-to-understand example. I also wonder whether the introduction of this concept is needed in the article. In my opinion not, therefore this section could also be deleted.
- If I understand correctly, after the preprocessing of the documents, a section-term matrix is resulted containing the TF-IDF values. If I’m right, I suggest calling this matrix a section-term matrix to clarify the resulted form of the preprocessing process. Furthermore, I also suggest presenting a small example from a section view.
- Please give the readers some information about the length of the resulted section-term vectors. How many terms were collected in these datasets?
- The base learners should be presented in more detail (lines between 231 and 239).
- Line 258: I suggest referring to Table 1 for datasets for the sake of clarity.
- Figures 5, 6, and 7 need to be more detailed evaluations.
Author Response
REVIEWER 1
The article proposes a novel multi-view ensemble learning architecture for structured text classification. The proposed methodology is well documented and the figures in the article are informative.
However, the proposed method is well-designed, the efficiency and usefulness of the proposed method need to be proven by evaluating more documents.
Main comments:
- The introduction does not give an overview of the aim of the proposed methodology and the possible application areas. Subsections “Ensemble Classifiers” and “Multi-view Ensemble Learning” present the related machine learning terms and areas at a high standard, but the aim and motivation of the research are entirely missing before these subparagraphs. Therefore, I suggest moving these subsections into a new Section (e.g. called “Theoretical background”) and inserting a brand new section (Introduction) before them containing the scope, the motivation of the research, and the possible application areas.
Thank you for your comments. They have undoubtedly helped to improve the article.
The suggested "Theoretical background" section has been incorporated, focusing the "Introduction" section on presenting the motivation for the research, the goal of the work and the possible areas of application.
- The main problem of the article arises from the methodology of the comparison of the results. If I understand correctly, the efficiency of the proposed method was compared to text classification results processing only on the Title and the Abstracts of the documents. It is not fair because the proposed method could process the whole document, but the baseline method only the title and the abstract. Therefore, it would be necessary to compare the results of the multi-view ensemble method with the result of other (not multi-view) methods processing the whole text of the documents.
Indeed, the explanation of the experiments performed was confusing and incomplete.
To evaluate the effectiveness of the proposed multi-view ensemble learning method to structured text classification, two different studies were carried out. Firstly, a comparison was between classical text classification on Title and Abstract, and classification using the proposed multi-view ensemble learning technique on full-text sections. Subsequently, a study was carried out on the efficiency of the model with respect to full-text classification techniques using a single classifier.
The previous paragraph has been included in the "Results and Discussion" section. In addition, two subsections have been created, one for each of the experiments performed. The second subsection collects the results of comparing the multi-view ensemble method with the use of a single classifier (namely, a SVM) processing the whole text of the documents.
Minor comments:
- Related works and their efficiency should be presented in more detail.
"Related works" section has been included, which details the works of other authors that are related to our proposal.
- Please clarify the terms “multi-view semi-supervised algorithms” through an easy-to-understand example. I also wonder whether the introduction of this concept is needed in the article. In my opinion not, therefore this section could also be deleted.
Following the reviewer's recommendations, the term "multi-view semi-supervised algorithms" has been deleted and the section has been considerably reduced.
- If I understand correctly, after the preprocessing of the documents, a section-term matrix is resulted containing the TF-IDF values. If I’m right, I suggest calling this matrix a section-term matrix to clarify the resulted form of the preprocessing process. Furthermore, I also suggest presenting a small example from a section view.
Thank you very much for the suggestion. The term Section-Term Matrix (line 260) has been included to increase clarity in the explanation of the preprocessing process.
On the other hand, in Subsection 5.2 we have incorporated a table (Table 2) with an example of term frequencies per section.
- Please give the readers some information about the length of the resulted section-term vectors. How many terms were collected in these datasets?
In Subsection 5.2 we have incorporated a table (Table 3) with the size of the vocabulary per section in the total OHSUMED corpus.
- The base learners should be presented in more detail (lines between 234 and 242).
The manuscript has been modified to indicate that SVMs are configured with a Radial Basis Function kernel with the default parameters provided by the WEKA environment
- Line 258: I suggest referring to Table 1 for datasets for the sake of clarity.
As indicated, Table 1 has been referenced in Figure 4 for clarity.
- Figures 5, 6, and 7 need to be more detailed evaluations.
Thanks for the suggestion. In the Discussion and Conclusions sections we have extended the reflections on the results of the different experiments. In addition, in this last section we have included future work that can improve the performance of the proposed model.
It should also be noted that the readability of the figures has been improved as can be seen in the new version of the article.
Reviewer 2 Report
Dear Authors,
The content of your article fits perfectly within the scope of Information journal's "Novel Methods and Applications in Natural Language Processing" Special Issue. One reason is that a key theme of the research involves issues of ensemble learning. This is very important for solutions to improve the task of text classification.
The Authors proposed a novel stacking technique with views corresponding to sections of the documents for full-text classification. Your research is at the same time pioneering from the viewpoint of using a multi-view ensemble learning scheme. It is relevant and interesting.
The above-mentioned goal was based on the 21 publications analysed in the Introduction.
The results of the study carried out on a biomedical dataset are given and explained. The authors proved that the approach proposed by them is much more effective.
The article contains some new data.
The paper is presented in logical way and overall written well.
The text is clear and easy to read.
The content of the Conclusions is consistent with the evidences and arguments presented and addresses the main question asked.
Comments and Suggestions for Authors
- The study considered the C4.5 decision tree as meta-learner. What other alternatives were there?
- The following typos were noted in the article: e.g., please:
i) to improve the readability of the contents in Figures 4-7,
ii) correcting the notation of decimal separators in Tables 1 and 2.
Author Response
REVIEWER 2
The content of your article fits perfectly within the scope of Information journal's "Novel Methods and Applications in Natural Language Processing" Special Issue. One reason is that a key theme of the research involves issues of ensemble learning. This is very important for solutions to improve the task of text classification.
The Authors proposed a novel stacking technique with views corresponding to sections of the documents for full-text classification. Your research is at the same time pioneering from the viewpoint of using a multi-view ensemble learning scheme. It is relevant and interesting.
The above-mentioned goal was based on the 21 publications analysed in the Introduction.
The results of the study carried out on a biomedical dataset are given and explained. The authors proved that the approach proposed by them is much more effective.
The article contains some new data.
The paper is presented in logical way and overall written well.
The text is clear and easy to read.
The content of the Conclusions is consistent with the evidences and arguments presented and addresses the main question asked.
Comments and Suggestions for Authors
- The study considered the C4.5 decision tree as meta-learner. What other alternatives were there?
In the manuscript we have included references to other works that use decision trees as meta-learners as well. As stated in line 276, we have tested Naive Bayes, C4.5 and SVM as base learners and meta-learners.
As a future work (this explanation has been included in the Conclusions Section), it would be optimal to automatically deduce which models are the best suited. Indeed, each corpus and each section should be treated with the most appropriate algorithm (base-learner), to later select a meta-classifier that outperforms the rest. However, we decided to go with the most general options possible to present the architecture.
- The following typos were noted in the article: e.g., please:
- i) to improve the readability of the contents in Figures 4-7,
Thanks for the suggestion. The readability of the figures has been improved as can be seen in the new version of the article.
- ii) correcting the notation of decimal separators in Tables 1 and 2.
The paper has been revised so that decimals are separated by a period, while the comma is reserved for separating thousands.
Round 2
Reviewer 1 Report
The authors followed my suggestions and have improved the paper. All suggestions were taken into account.